# *Lepidium virginicum* Water-Soluble Chlorophyll-Binding Protein with Chlorophyll A as a Novel Contrast Agent for Photoacoustic Imaging

**DOI:** 10.3390/s25113492

**Published:** 2025-05-31

**Authors:** Victor T. C. Tsang, Hannah H. Kim, Bingxin Huang, Simon C. K. Chan, Terence T. W. Wong

**Affiliations:** Department of Chemical and Biological Engineering, Hong Kong University of Science and Technology, Hong Kong 999077, China; tcvtsang@connect.ust.hk (V.T.C.T.); hkimam@connect.ust.hk (H.H.K.); bhuangao@connect.ust.hk (B.H.); ckchanbq@connect.ust.hk (S.C.K.C.)

**Keywords:** photoacoustic imaging, contrast agent, chlorophyll-binding protein

## Abstract

**Highlights:**

**What are the main findings?**

**What is the implication of the main finding?**

**Abstract:**

Photoacoustic (PA) imaging (PAI) holds great promise for non-invasive biomedical diagnostics. However, the efficacy of current contrast agents is often limited by photobleaching, toxicity, and complex synthesis processes. In this study, we introduce a novel, biocompatible PAI contrast agent: a recombinant water-soluble chlorophyll-binding protein (WSCP) from *Lepidium virginicum* (LvP) reconstituted with chlorophyll a (LvP-chla). LvP-chla exhibits a strong and narrow absorption peak at 665 nm, with a molar extinction coefficient substantially higher than oxyhemoglobin and deoxyhemoglobin, enabling robust signal generation orthogonal to endogenous chromophores. Phantom studies confirmed a linear relationship between PA signal amplitude and LvP-chla concentration, demonstrating its stability and reliability. In vitro cytotoxicity testing using 4T1 cells showed high cell viability at 5 mg/mL, justifying its use for in vivo studies. In vivo experiments with a 4T1 tumor-bearing mouse model demonstrated successful tumor localization following intratumoral injection of LvP-chla, with clear visualization via spectroscopic differentiation from endogenous absorbers at 665 nm and 685 nm. Toxicity assessments, both in vitro and in vivo, revealed no adverse effects, and clearance studies confirmed minimal retention after 96 h. These findings show that LvP-chla is a promising contrast agent that enhances PAI capabilities through its straightforward synthesis, stability, and biocompatibility.

## 1. Introduction

Photoacoustic (PA) imaging (PAI) has emerged as a powerful non-invasive imaging modality that synergistically combines high molecular contrast with optical excitation and deep tissue penetration with ultrasound detection [1,2]. By relying on the absorption of pulsed laser light by chromophores inside biological tissues, the absorbers generate ultrasonic waves through transient thermoelastic expansion, enabling the visualization of anatomical structures and the provision of functional information at significant depths beyond the optical limit with ballistic photons. Endogenous chromophores such as hemoglobin and melanin inherently provide contrast for PAI [2,3]. However, the limited specificity and potential overlap of their absorption spectra often restrict the sensitivity and specificity required for targeted imaging applications.

Consequently, developing exogenous contrast agents that exhibit strong absorption in the far-red to near-infrared (NIR) window (650–900 nm) has become a focal point in advancing PAI technologies. An ideal contrast agent for imaging should possess high molar extinction coefficients, biocompatibility, stability, and efficient clearance from the body [4,5]. Various materials, including organic dyes, nanoparticles, and recombinant proteins, have been explored as PA contrast agents. Organic dyes like indocyanine green solution (ICG) suffer from photobleaching and aggregation at high concentrations, limiting their stability and imaging consistency [4]. Nanoparticles, while offering strong NIR absorption, often exhibit potential toxicity and complex synthesis processes, hindering clinical translation [6]. Genetically encodable reporters, such as phytochromes, rely on endogenous cofactors like biliverdin, which vary in availability across tissues and may require co-expression of heme oxygenase for optimal performance [7]. In contrast, the recombinant water-soluble chlorophyll-binding protein from *Lepidium virginicum* reconstituted with chlorophyll a (LvP-chla) offers a high molar extinction coefficient (59,716 M^−1^cm^−1^ at 665 nm), exceptional photo- and thermostability, straightforward synthesis using naturally abundant chlorophyll a from spinach, and independence from endogenous cofactors, making it a promising candidate to overcome these limitations [8,9,10,11].

As a result, our study addresses these challenges by introducing LvP-chla as a novel PA contrast agent [12]. Chla, the primary light-harvesting pigment in plants, can be readily sourced from stable materials such as spinach leaves, facilitating external reconstitution without reliance on endogenous factors. LvP-chla can be synthesized using exogenously supplied and naturally occurring pigments. Additionally, LvP-chla is reported to possess exceptional photo- and thermostability, with its tetrameric structure resisting dissociation and denaturation even at 100 °C [13,14,15,16]. These properties are particularly favorable for PA contrast agents, where PA signal generation involves laser illumination and thermoelastic expansion, processes that can induce thermal and photochemical stress. We hypothesize that LvP-chla can serve as a biocompatible and efficient contrast agent for PAI, overcoming the limitations of current contrast agents. To validate this hypothesis, here, we characterize the optical properties of LvP-chla, evaluate its PA signal generation efficiency in vitro and in vivo, and assess its spectroscopic differentiation from endogenous absorbers. Additionally, we conducted toxicity studies to ensure its safety for potential clinical applications. Our findings suggest that LvP-chla not only enhances PA imaging performance but also holds significant potential for advancing non-invasive diagnostic imaging.

## 2. Materials and Methods

### 2.1. Contrast Agent Synthesis

#### 2.1.1. Gene Construction and Expression of LvP

The gene encoding the WSCP from LvP (PDB code: 6GIW), reported by Palm et al. [15], was synthesized and cloned into the pET22b expression vector. *Escherichia coli* (*E. coli*) strain BL21(DE3) carrying the pET22b::LvWSCP construct was cultured in Luria–Bertani medium supplemented with 100 mg/L ampicillin at 37 °C with shaking until the optical density at 600 nm (OD_600_) reached 0.6–1.2. Protein expression was induced by adding 400 μM isopropyl β-D-1-thiogalactopyranoside and incubating at 37 °C for 3 h.

#### 2.1.2. Reconstitution of LvP with Chla

Chla was reconstituted into LvP following the method reported by Li et al. [17]. Fresh spinach leaves were used as the source of chla. The induced *E. coli* pellets containing LvP were homogenized with spinach leaves at a mass ratio of approximately 5:1 (spinach/*E. coli* pellet) in lysis buffer containing 300 mM sodium chloride, 20 mM Tris buffer, and 0.1 mM phenylmethylsulfonyl fluoride at pH 7.5. The homogenate was sonicated using a Branson sonicator at 150 W power with a duty cycle of 50% (0.5 s on, 0.5 s off) for a total of 10 min, divided into two 5-min sessions with a 60-s rest interval between each session. This method facilitates the incorporation of chla into the LvP protein, generating the desired contrast agent, LvP-chla, for PAI.

#### 2.1.3. Protein Purification

Following sonication, the homogenate was centrifuged at 14,000× *g* for 30 min at 4 °C to separate the supernatant containing the reconstituted LvP-chla from insoluble fractions. The supernatant was purified using standard nickel–nitrilotriacetic acid (Ni-NTA) affinity chromatography with HisTrap™ HP columns (Cytiva, Marlborough, MA, USA). The purification process involved washing with buffer (300 mM NaCl, 20 mM Tris, 25 mM imidazole, pH 7.5) to remove impurities and eluting the purified protein with buffer containing 300 mM NaCl, 20 mM Tris, and 500 mM imidazole at pH 7.5. The purified LvP-chla further underwent buffer exchange and concentration using Amicon Ultra centrifugal filters (Sigma Aldrich, St. Louis, MO, USA). The final LvP-chla solution was prepared in phosphate-buffered saline (PBS) for injection. For storage, the protein solution was flash-frozen in liquid nitrogen and maintained at −80 °C until further use.

#### 2.1.4. Spectral Measurement

The absorption spectrum of the protein solution was measured with a Varioskan LUX multimode microplate reader with PBS as the blank reference. Spectral scans were performed over a wavelength range from 300 nm to 900 nm with 1 nm resolution. After that, the concentration of the protein solution was determined by a NanoDrop spectrophotometer (Thermo Fisher Scientific, Waltham, MA, USA). The molar extinction coefficient was then calculated using the Beer–Lambert law with the normalized absorbance and the concentration. The photoluminescence quantum yield (PLQY) was recorded on an ultraviolet to NIR absolute PLQY spectrometer (Quantaurus-QY, Hamamatsu Photonics K.K., Shizuoka, Japan) at room temperature.

### 2.2. PAI Setup

The PAI system employed in this study is based on the configuration described by Wang et al. [18]. Minor modifications were implemented to adopt the system for small animal imaging (Figure 1a). The imaging setup consists of a linear array ultrasound transducer (UST) positioned coaxially with the optical fiber bundle using a custom-made 3D-printed adapter (Figure 1b) within a water tank. An optically transparent membrane was adhered to the bottom of the water tank to facilitate effective coupling of the generated PA signals. The tumor-bearing mouse was placed in a supine position. Ultrasound gel was applied directly to the skin of the nude mouse to enhance acoustic coupling between the tissue and the UT. The optical fiber bundle delivers pulsed laser excitation from an optical parametric oscillator (OPO; NT352C-20-FWS-SH-H, EKSPLA Inc., Vilnius, Lithuania) to the imaging region. Data acquisition was managed by a commercial system (AcousticX, Cyberdyne Inc., Tsukuba, Japan) connected to a personal computer (PC), as shown in Figure 1c.

### 2.3. Phantom Imaging

Chlorophyll-reconstituted WSCP proteins dissolved in PBS were serially diluted to create a concentration gradient from 1 mg/mL to 10 mg/mL. These protein solutions were carefully loaded into thin capillary tubes via capillary action to create phantom samples for PAI. Additionally, 1 mL of mouse whole blood was harvested via cardiac puncture, and 0.1 mL of ethylenediamine tetraacetic acid solution was added to prevent coagulation. A 0.5× diluted blood sample was prepared by mixing 0.5 mL of whole blood with 0.5 mL of PBS and loaded into separate capillary tubes. Both ends of the loaded capillary tubes were sealed with glue and gently passed over an open flame to ensure the containment of the samples. The prepared capillary tubes were placed on a stable holder within the imaging field to facilitate the acquisition of PA signals generated by the contrast agents and blood samples under laser excitation.

### 2.4. Animal Imaging

#### 2.4.1. Tumor Mouse Model

Nu/J mice (The Jackson Laboratory, Bar Harbor, ME, USA), aged 4–5 weeks, were utilized to establish 4T1 cell line allografts. Each mouse received subcutaneous injections of 1 × 10^6^ 4T1 cells suspended in 100 µL PBS at two sites: the right hindlimb and the left forelimb. Mice were housed under standard conditions (temperature: ~22 °C, humidity: 40–70%, 12 h light/dark cycles) with free access to sterile food and water. Tumor growth was monitored regularly, and imaging experiments were conducted once tumor volumes reached approximately 2000 mm^3^. All experiments were carried out in conformity with a laboratory animal protocol approved by the Health, Safety, and Environment Office of The Hong Kong University of Science and Technology (Approval Number: AEP-2022-0010).

#### 2.4.2. Image Acquisition

The tumor-bearing mice were sedated via intraperitoneal injection of a ketamine/xylazine/saline (KXS) cocktail composed of 17.5% *v*/*v* ketamine, 2.5% *v*/*v* xylazine, and 80% *v*/*v* sterile saline at a dosage of 5 µL/g body weight. The plane of anesthesia was confirmed by performing a toe-pinch reflex test. The mouse was then positioned in a supine orientation on the imaging stage. A total of 200 µL of 5 mg/mL LvP-chla solution was administered via intratumoral injection. Ultrasound gel was applied to the tumor site and surrounding region to facilitate acoustic coupling. PAI was performed using a laser wavelength of 665 nm, corresponding to the absorption maximum of LvP-chla, to acquire volumetric PA images along the midline towards the caudal direction. Subsequently, the same volumetric scan was repeated using a 685 nm laser excitation, at which the molar extinction coefficient of LvP-chla is reduced by half. The laser operated at a repetition rate of 20 Hz with pulse widths ranging from 3 to 5 ns. The laser fluence on the tissue surface was approximately 16.1 mJ/cm^2^ for 665 nm and 15.6 mJ/cm^2^ for 685 nm, both below the American National Standards Institute safety limit of 20 mJ/cm^2^.

### 2.5. Toxicity Test

#### 2.5.1. Cytotoxicity Assay

The cytotoxicity of LvP-chla was assessed using the CyQUANT MTT Cell Viability Asssay kit (Thermo Fisher Scientific, Waltham, MA, USA) with 4T1 cells. The 4T1 cells were seeded at a density of 3 × 10^4^ cells per well in 96-well plates and cultured in RPMI-1640 medium supplemented with 10% fetal bovine serum and 1% penicillin-streptomycin at 37 °C in a 5% CO_2_ atmosphere. After 24 h, cells were treated with fresh medium mixed with LvP-chla at concentrations of 0, 1, 3, 5, 7, and 10 mg/mL in 4 replicates. A negative control (medium with MTT reagent without cells) and a positive control (untreated cells, 0 mg/mL LvP-chla) were included. After 12 h of incubation, 10 µL of 12 mM MTT reagent was added to each well, and the plates were incubated for 4 h at 37 °C. The resulting formazan crystals were solubilized by adding 100 µL of the kit’s SDS-HCl solution (0.1% SDS in 0.01 M HCl) to each well, then incubating for 10 min at 37 °C with gentle mixing. Absorbance was measured at 570 nm using a Varioskan LUX multimode microplate reader (Thermo Fisher Scientific, Waltham, MA, USA). Cell viability was calculated as a percentage relative to the positive control after subtracting the negative control absorbance. Data were analyzed using one-way ANOVA with Dunnett’s post-hoc test to compare treated groups to the positive control, with significance set at *p* < 0.05. GraphPad Prism (version 9.0) was used for statistical analysis.

#### 2.5.2. Histology Study

To evaluate the potential cytotoxicity of LvP-chla in vivo, key organs (e.g., liver and kidney) and tumor tissues were harvested from euthanized mice following 96 h post-injection of 5 mg/mL LvP-chla. Euthanasia was performed by overdose of the KXS cocktail as described above. Harvested tissues were immediately fixed in 4% neutral-buffered formalin at room temperature for 24 h. After fixation, tissues were processed using a tissue processor (Revos, Thermo Fisher Scientific, Waltham, MA, USA) for 12 h, followed by paraffin embedding. Paraffin-embedded tissues were sectioned at a thickness of 5 µm using a microtome (RM2235, Leica Microsystems, Wetzlar, Germany) and mounted onto glass slides. The tissue sections were stained with hematoxylin and eosin (H&E), and the stained slides were imaged using a digital slide scanner (NanoZoomer-SQ, Hamamatsu Photonics K.K., Shizuoka, Japan).

## 3. Results

LvP-chla exhibits a characteristic chla Qy absorption band at 665 nm (Figure 2a), with a molar extinction coefficient (ε) of 59,716 M^−1^cm^−1^. This value is significantly higher than that of hemoglobin (Hb) and oxyhemoglobin (HbO_2_) at the same wavelength, as shown in the comparative spectra (Figure 2b,d). The absorbance at 653 nm and 685 nm is reduced by half, resulting in a narrow Qy band with a full width at half maximum of 20 nm. The excitation spectrum monitored at 672 nm emission (Figure 2c) reveals multiple excitation peaks, with maxima at 437 nm, 619 nm, and 663 nm. The PLQY of the LvP-chla was measured at 7.5%, which is lower than the 29% reported by Kotewicz et al. for free chla dissolved in ethanol [14].

Phantom studies were conducted to evaluate the PA signal generation efficiency of LvP-chla (Figure 3). LvP-chla successfully generated PA signals across concentrations ranging from 1 mg/mL to 10 mg/mL under 665 nm excitation. The selection of 665 nm and 685 nm excitation wavelengths leverages the maximum absorption peak and halves the absorption of the LvP-chla spectral profile while enabling differential imaging with minimal penetration depth variations and consistent laser energy output owing to their narrow separation. A linear relationship between PA signal amplitude and LvP-chla concentration was observed for both 665 nm and 685 nm excitations (Figure 3a), with whole blood and 0.5× blood included as controls. The linear regression analysis yielded high coefficients of determination (R^2^ = 0.9905 and 0.9079 for 665 nm and 685 nm excitation, respectively) with regression equations of y = 17.40x − 5.818 and y = 6.104x − 14.29, demonstrating robust concentration-dependent PA signal generation. The PA amplitude exhibited a threefold reduction at 685 nm compared to 665 nm. For the penetration depth study, 5 mg/mL LvP-chla was selected as it provides sufficient signal strength while maintaining low viscosity and cytotoxicity, making it suitable for subsequent intratumoral injection. The penetration depth characteristics were evaluated using chicken breast tissue as a biological scattering medium (Figure 3c). Quantitative signal analysis revealed the half-maximum amplitude at 6.30 mm depth (Figure 3d), indicating promising potential for deep-tissue imaging applications.

In order to evaluate the tolerable dosage for in vivo imaging, the cytotoxic effects of LvP-chla on 4T1 cells were evaluated using the MTT assay. A clear concentration-dependent cytotoxic response was observed after 12-h exposure to LvP-chla (Figure 4). LvP-chla treatments at lower concentrations (1, 3, and 5 mg/mL) showed no significant cytotoxic effects, maintaining cell viability close to the untreated control level. Conversely, higher LvP-chla concentrations (7 and 10 mg/mL) significantly reduced cell viability compared to the untreated control (*p* < 0.05 and *p* < 0.01, respectively). These results demonstrate that LvP-chla concentrations at or above 7 mg/mL exert pronounced cytotoxic effects on 4T1 cells, while 5 mg/mL is a safe and effective dose for in vivo studies.

Following successful in vitro validation, the efficacy of LvP-chla as an in vivo PA contrast agent was evaluated using a nude mouse 4T1 allograft model. Utilizing the optimized 5 mg/mL concentration from the cytotoxicity test, 200 μL of LvP-chla was administered intratumorally, followed by dual-wavelength spectroscopic PAI at 665 nm and 685 nm (Figure 5). The experimental workflow is illustrated in Figure 5a. Baseline imaging prior to LvP-chla administration revealed typical endogenous PA signals at both wavelengths (Figure 5d,e) with negligible contrast in the differential images at the tumor site (Figure 5f). The concurrent ultrasound imaging (Figure 5c) provides essential anatomical context, enabling precise tumor localization and structural information while demonstrating the system’s dual-modality capability. Post-administration imaging (Figure 5g–k) demonstrated pronounced PA signal enhancement localized within the tumor region, with ultrasound images (Figure 5h) continuing to provide anatomical guidance for accurate spatial registration of the PA image. The differential imaging approach (Figure 5k) particularly enhanced visualization of the contrast agent distribution by effectively eliminating background signals while maintaining the structural context provided by ultrasound imaging. There is a 6.52-fold increase in mean PA signal strength in the tumor area, as shown in Figure 5l in the differential image, driven by LvP-chla’s strong absorption at 665 nm (59,716 M^−1^cm^−1^) and effective background subtraction, compared to a 1.54-fold increase at 665 nm and a 1.24-fold increase at 685 nm. These findings highlight the capability of LvP-chla, in conjunction with dual-wavelength differential PAI, to provide high-contrast molecular information overlaid on ultrasound images, representing a significant advancement in dual-modality molecular imaging that combines the specificity of PAI with the anatomical information of ultrasound.

Another important criterion for evaluating contrast agents is the absence of undesired long-term retention. To assess in vivo clearance, longitudinal photoacoustic (PA) imaging was conducted after LvP-chla injection in nude mice bearing 4T1 tumors, using pre-injection PA signals as baseline references. Differential PA images acquired over 96 h clearly showed a gradual decrease in signal intensity (Figure 6). Exponential decay analysis of normalized PA signals revealed signal half-lives (t½) of approximately 11.6 h (R^2^ = 1.000) with signal plateauing 24 h post-injection. Importantly, PA signals returned to baseline within 96 h post-injection, indicating minimal long-term retention of LvP-chla in tumor tissue.

For the toxicity check of the contrast agent, the mice exhibited no signs of distress throughout the observation period, and tumor size remained unchanged. Histological analysis of vital organs (Figure 7) reveals no cellular damage or abnormality in the heart (Figure 7a), kidneys (Figure 7d), or liver (Figure 7e) following LvP-chla administration. While splenomegaly was observed with an increase in white blood cells in the spleen, this is likely attributed to an immune response towards tumor growth rather than LvP-chla injection, as the normal white pulp is observed in Figure 7b. Overall, these results demonstrate that LvP-chla exhibits no observable toxicity, enabling safe and longitudinal monitoring of targeted biomarkers in deep tissues.

## 4. Discussion

Genetically encodable photoacoustic (PA) reporters have emerged as promising contrast agents, offering versatility through synthetic biology approaches that enable targeted ligand integration [8]. Most genetically encodable probes have relied on bacterial phytochromes such as BphP1 derivatives, stabilizing near-infrared (NIR) absorbing cofactors like biliverdin. However, these approaches depend on endogenous cofactors, limiting their reliability across varying tissues and organisms. Strategies involving co-expression of heme oxygenase genes have been employed to overcome this limitation by enhancing cofactor availability [19,20].

In this study, we demonstrated that the LvP-chla complex exhibits favorable optical and functional properties, highlighting its potential as a robust PA contrast agent. LvP-chla’s exceptionally high molar extinction coefficient (59,716 M^−1^cm^−1^ at 665 nm) surpasses endogenous absorbers such as hemoglobin, enabling enhanced PA signal detection. Moreover, the narrow Qy absorption band (~20 nm FWHM) facilitates spectroscopic differentiation from background tissue chromophores, addressing a common challenge in PA imaging. Additionally, LvP-chla maintains optical stability across a wide concentration range, evidenced by a linear relationship between PA signal amplitude and concentration at both 665 nm and 685 nm excitation wavelengths (R^2^ = 0.9905 and 0.9079, respectively). This linearity, combined with robust signal generation, contrasts with common small-molecule NIR absorbers, such as indocyanine green, which typically aggregate at higher concentrations, complicating their practical use [21,22]. In comparison, existing PA contrast agents like ICG, which suffer from photobleaching and aggregation, or phytochrome-based reporters that depend on variable endogenous cofactors, LvP-chla’s straightforward synthesis, high stability, and biocompatibility provide a significant advantage for reliable and translatable PAI applications.

An intriguing observation in this study was that the measured PA signal ratio between 665 nm and 685 nm excitation wavelengths (approximately 3-fold difference) exceeded the absorbance ratio (approximately 2-fold). Several potential explanations exist for this discrepancy, including wavelength-dependent variations in the Grüneisen parameter or local microenvironmental effects influencing heat generation efficiency. Moreover, the observed reduction in PLQY of LvP-chla compared to free chlorophyll a could enhance non-radiative decay pathways and heat generation, thereby potentially increasing PA efficiency. While these interesting findings warrant further investigation, such studies are beyond this current paper’s scope, which aims to establish a clear proof-of-concept for LvP-chla as a viable PA contrast agent.

In vivo experiments further validated LvP-chla’s imaging performance. Successful tumor visualization post-intratumoral injection, coupled with clear spectral distinction from endogenous absorbers, underscores its potential for targeted imaging. Unlike typical BphP derivatives that employ dimerization, LvP-chla reportedly forms tetramers with chlorophyll molecules, which theoretically enhances signal amplification and may improve PA imaging sensitivity and specificity [23,24]. Moreover, the achieved imaging depth (~6.3 cm in scattering tissue) is promising for clinical translation.

Toxicity assessments are critical in evaluating clinical viability. LvP-chla demonstrated no observable toxicity or distress in experimental mice, supporting its biocompatibility. Notably, the transient nature of PA signals, returning to baseline within 96 h post-injection, indicates efficient systemic clearance—a desirable trait for clinical contrast agents. Additionally, while systemic administration via intravenous injection would broaden LvP-chla’s clinical applicability, it could introduce complexities related to immunogenicity and interactions with serum proteins. Incorporating targeting ligands, such as affibodies or antibodies, could enhance the tumor-specific accumulation of LvP-chla during intravenous administration, facilitating its translation to clinical applications requiring non-invasive delivery [25,26].

Despite these promising results, certain limitations should be acknowledged. LvP-chla’s absorption peak in the far-red region (~665 nm) may limit imaging depth compared to NIR wavelengths. Additionally, the structural flexibility of LvP to bind various tetrapyrroles structurally similar to chla presents opportunities for developing novel PA contrast agents [27]. Future research could explore integrating pigments such as bacteriochlorophyll a to shift absorption peaks further into the NIR region [28,29,30].

Overall, this study provides a solid proof-of-concept demonstrating LvP-chla’s strong potential as a robust, biocompatible, and easily synthesized PA contrast agent. Future investigations addressing these open questions will further enhance LvP-chla’s clinical potential and broaden its translational applicability.

## 5. Conclusions

This study presents LvP-chla, a recombinant chlorophyll-binding protein reconstituted with chla, as a novel and effective contrast agent for PAI. LvP-chla exhibits strong far-red optical absorption with a high molar extinction coefficient and a narrow Qy absorption band, enabling robust and distinguishable PA signals. It demonstrates concentration-dependent signal generation, enhanced imaging contrast, and effective tumor site visualization in both phantom and in vivo experiments using a 4T1 tumor model. Dual-wavelength differential imaging at 665 nm and 685 nm further distinguished LvP-chla from endogenous absorbers, while penetration depth studies confirmed its potential for deep-tissue imaging (up to ~6.30 mm) in scattering media.

Toxicity evaluations confirmed high biocompatibility, revealing no adverse effects at effective imaging concentrations and efficient systemic clearance within 96 h post-injection. These results strongly support LvP-chla’s potential for clinical translation as a safe, stable, and reliable PA contrast agent.

While intriguing observations such as the quantum yield discrepancy, wavelength-dependent PA signal differences, and clearance kinetics require further investigation, this initial proof-of-concept study demonstrates LvP-chla’s significant potential for enhancing PAI applications. Future research on optimizing optical properties, shifting absorption peaks toward the NIR region, and investigating systemic administration routes will further strengthen its clinical applicability.

In conclusion, LvP-chla represents an exciting and valuable addition to the growing toolkit of PA contrast agents, promising substantial advancements in non-invasive biomedical imaging and diagnostics.

## 6. Patents

V. T. C. T., H. H. K., and T. T. W. W. have applied for a patent (US Provisional Patent Application No.: 63/686,861) related to the work reported in this manuscript.

## Figures and Tables

**Figure 1 sensors-25-03492-f001:**
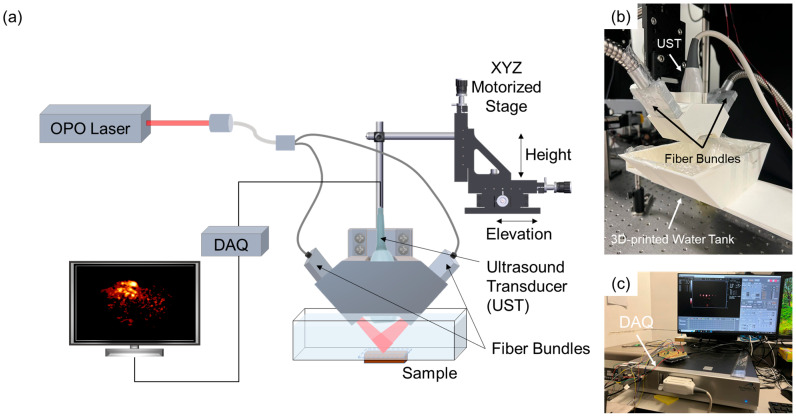
The PAI setup. (**a**) Schematic of the PAI system. (**b**) Photograph of the 3D-printed adapter for the two fiber bundles and UST on top of the water tank wrapped with an optically transparent membrane. (**c**) Photograph of the DAQ and the PC-integrated commercial system (AcousticX, Cyberdyne Inc.). OPO, optical parametric oscillator; DAQ, data acquisition system.

**Figure 2 sensors-25-03492-f002:**
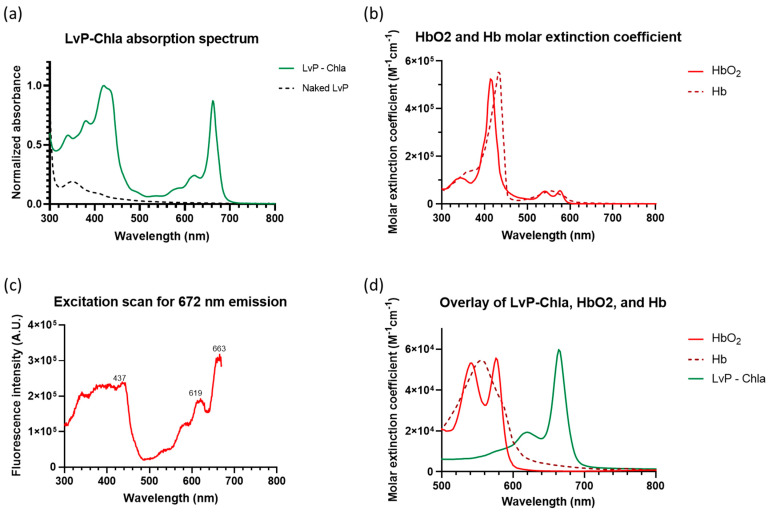
Spectroscopic characterization of LvP-chla. (**a**) UV–vis absorption spectra of LvP-chla (solid green line) and naked LvP (dashed black line), showing the characteristic chlorophyll a absorption peaks. (**b**) Molar extinction coefficient spectra of oxyhemoglobin (HbO_2_, solid red line) and deoxyhemoglobin (Hb, dashed red line) in the visible to near-infrared region. (**c**) Excitation spectrum of LvP-chla monitored at 672 nm emission, revealing excitation maxima at 437 nm, 619 nm, and 663 nm. (**d**) Overlay of molar extinction coefficient spectra of LvP-chla (green line), HbO_2_ (solid red line), and Hb (dashed red line), demonstrating the superior absorption of LvP-chla at 665 nm compared to hemoglobin species.

**Figure 3 sensors-25-03492-f003:**
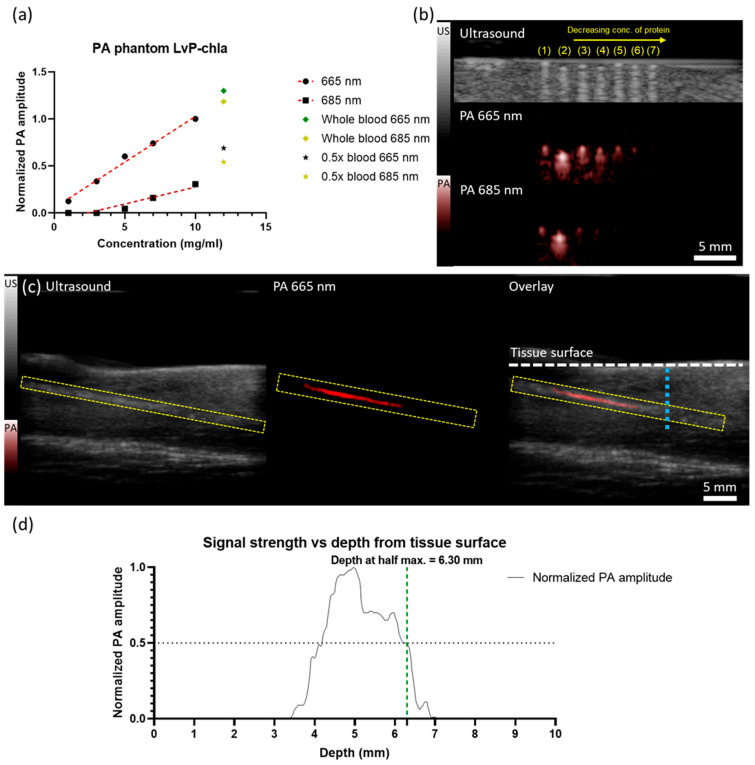
Photoacoustic characterization of LvP-chla and blood phantoms. (**a**) Linear regression analysis showing normalized PA amplitude response of LvP-chla at different concentrations under 665 nm (red dots) and 685 nm (yellow dots) excitation, with blood controls. (**b**) Dual-modality imaging of phantom samples showing ultrasound image (top), PA image at 665 nm (middle), and PA image at 685 nm (bottom). Samples shown are (1) 0.5x blood, (2) whole blood, and LvP-chla at decreasing concentrations: (3) 10 mg/mL, (4) 7 mg/mL, (5) 5 mg/mL, (6) 3 mg/mL, and (7) 1 mg/mL. (**c**) Depth penetration study showing ultrasound image (left), PA image at 665 nm (middle), and overlay (right) of a LvP-chla phantom (5 mg/mL) embedded obliquely in chicken breast tissue. Yellow dashed line boxes indicate the phantom position and white dashed line marks the tissue surface. (**d**) Quantitative analysis of PA signal strength versus depth from the tissue surface (cyan dotted line in right subfigure of figure (**c**)), showing imaging depth capability with the half-maximum signal at 6.30 mm depth.

**Figure 4 sensors-25-03492-f004:**
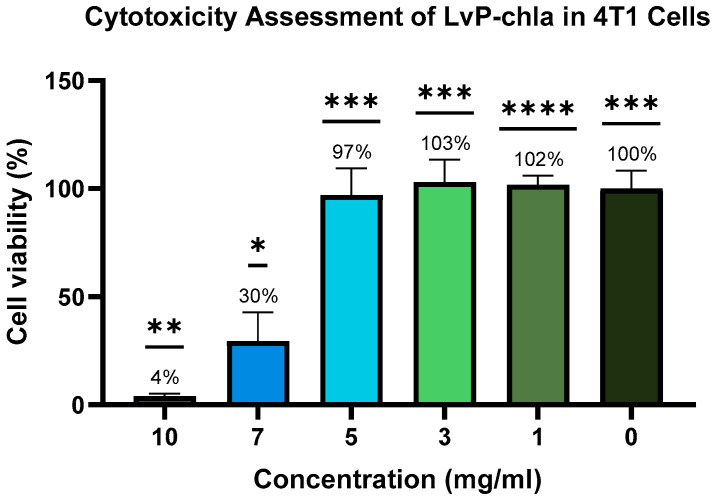
Cytotoxicity of LvP-chla assessed by CyQUANT MTT Cell Viability Assay in 4T1 cells. Cells were treated with LvP-chla at 0, 1, 3, 5, 7, and 10 mg/mL for 12 h. Cell viability was quantified as a percentage relative to untreated control cells (0 mg/mL LvP-chla) after subtracting background absorbance (negative control: medium with MTT reagent, no cells). Data represent mean ± standard deviation (*n* = 4). Statistical analysis was performed using one-way ANOVA followed by Dunnett’s post-hoc test (* *p* < 0.05, ** *p* < 0.01, *** *p* < 0.001, and **** *p* < 0.0001 versus positive control).

**Figure 5 sensors-25-03492-f005:**
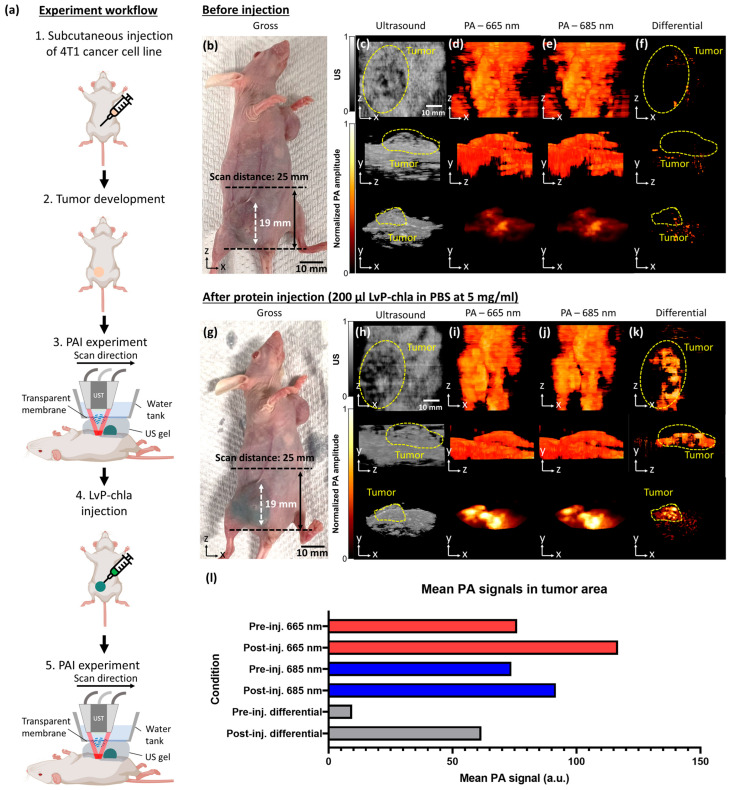
Experimental design and dual-modality imaging of LvP-chla delivery in a 4T1 tumor model. (**a**) Schematic workflow showing the experimental procedure: (1) subcutaneous injection of 4T1 cancer cells, (2) tumor development period, (3) initial PAI scanning, (4) intratumoral injection of LvP-chla, and (5) follow-up PAI experiment. Pre-injection imaging data (**b**–**f**): (**b**) Gross photograph of the tumor-bearing mouse, (**c**) ultrasound images showing tumor morphology in three orthogonal views, (**d**) PA images at 665 nm, (**e**) PA images at 685 nm, and (**f**) differential PA images (665 nm–685 nm). Post-injection imaging data (**g**–**k**): (**g**) Gross photograph after injection of 200 μL LvP-chla (5 mg/mL in PBS), (**h**) ultrasound images, (**i**) PA images at 665 nm, (**j**) PA images at 685 nm, and (**k**) differential PA images showing protein distribution. (**l**) Bar chart showing mean PA signals in the tumor region, averaged across xy, xz, and yz projections, before and after LvP-chla injection at 665 nm, 685 nm, and differential imaging. Yellow dashed lines indicate tumor boundaries. Scale bars: 10 mm for gross images.

**Figure 6 sensors-25-03492-f006:**
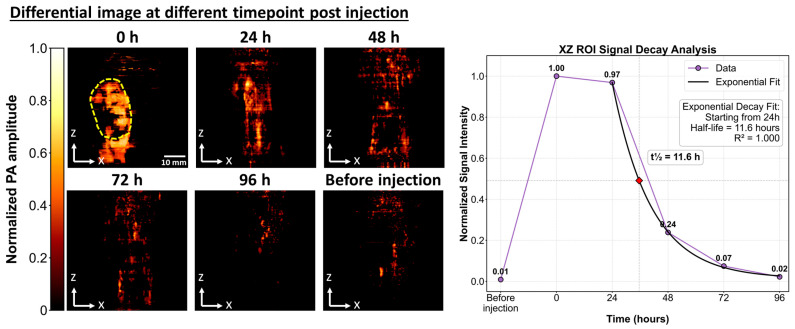
Spatiotemporal analysis of differential photoacoustic (PA) imaging showing protein distribution and clearance over time. xz-projection images before injection and at indicated time points (0, 24, 48, 72, and 96 h post-injection). Normalized PA signals within regions of interest (yellow dashed outlines at 0 h) were fitted to exponential decay models, yielding signal half-lives (t½) of 11.6 h (R^2^ = 1.000). Scale bars: 10 mm.

**Figure 7 sensors-25-03492-f007:**
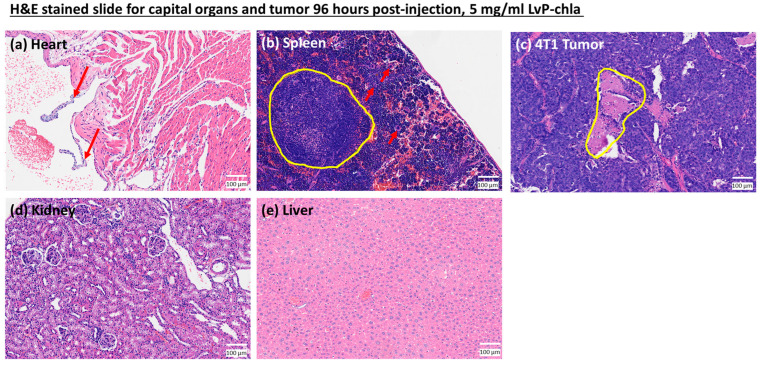
H&E stained tissue sections from mice treated with LvP-chla. (**a**) Heart, showing right atrium valves (red arrows); (**b**) Spleen, highlighting nominal white pulp (yellow circle) and megakaryocytes (red arrows); (**c**) 4T1 Tumor, demonstrating a necrotic region (yellow circle); (**d**) Kidney, displaying a normal glomerulus; (**e**) Liver, exhibiting normal histological morphology.

## Data Availability

The original contributions presented in this study are included in the article.

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
