# Peer review of "Lepidium virginicum Water-Soluble Chlorophyll-Binding Protein with Chlorophyll A as a Novel Contrast Agent for Photoacoustic Imaging"

_sensors, 2025, doi:10.3390/s25113492_

Round 1

Reviewer 1 Report

Comments and Suggestions for Authors

This study introduced a novel photoacoustic imaging (PAI) contrast agent, Lepidium virginicum water-soluble chlorophyll-binding protein reconstituted with chlorophyll a (LvP-chla). Through phantom studies, in vitro and in vivo testing, LvP-chla showed its low toxicity and biocompatibility, which demonstrated its potential as a stable and clinically translatable contrast agent for PAI. The experimental results are novel and very interesting, however the authors need to add some contents to make the paper more convincing:

1.In the introduction or discussion section, the introduction of the existing PA contrast agent and the comparison between it and the LvP-chla can be added to clarify the advantages of LvP-chla.

2.In Figure 5, quantitative statistical data of PA signals in the tumor area before and after LvP-chla injection can be added, which can display the enhancement effect of PA signals more obviously.

3.This paper only presented intratumoral injection experiments. What would be the result if the intravenous injection method was used? This will limit the application of this contrast agent in clinical translation.

4.In the discussion section, the author should add the discussion on how to modify its structure to move its absorption peak to the near-infrared window (700-900 nm), which will enhance its penetration depth for deeper imaging.

Author Response

Comment 1: In the introduction or discussion section, the introduction of the existing PA contrast agent and the comparison between it and the LvP-chla can be added to clarify the advantages of LvP-chla

Response: Thank you for the suggestion. We agree that comparing LvP-chla with existing PA contrast agents is crucial to highlight its novelty. We have added a paragraph in the Introduction (Section 1, Page 2, second paragraph) discussing common PA agents, their limitations, and LvP-chla’s advantages. A sentence in the Discussion (Section 4, Page 12, second paragraph) reinforces this comparison, summarizing how LvP-chla addresses these limitations, which are supported by our results.

Comment 2: In Figure 5, quantitative statistical data of PA signals in the tumor area before and after LvP-chla injection can be added, which can display the enhancement effect of PA signals more obviously.

Response: We appreciate your suggestion to include quantitative data to demonstrate the PA signal enhancement effect of LvP-chla. To address this, we have included a bar chart in a new panel (Figure 5(l)) showcasing the mean PA signal strength in the tumor region, averaged across xy, xz, and yz projections, before and after intratumoral injection at 665 nm, 685 nm, and in differential images. The Results section (Section 3) has been modified to highlight the signal enhancement, particularly with the differential imaging approach, which shows a 6.52-fold increase in the tumor area post-injection. The Figure 5 caption has been updated to describe the new panel. This addition strengthens the evidence for LvP-chla’s imaging efficacy and directly addresses your request.

Comment 3: This paper only presented intratumoral injection experiments. What would be the result if the intravenous injection method was used? This will limit the application of this contrast agent in clinical translation.

Response: Thank you for highlighting the potential of intravenous injection to enhance LvP-chla’s clinical applicability. In this study, we deliberately chose intratumoral injection to establish a robust proof-of-concept for LvP-chla’s efficacy as a PA contrast agent, focusing on its ability to generate strong, specific signals in a controlled tumor environment. Intravenous administration, while a valuable approach for clinical translation, was beyond the scope of this initial study, as our primary objective was to validate LvP-chla’s performance in a targeted setting. To address the potential for systemic delivery, we have revised the Discussion (Section 4, 5th paragraph) to include a statement that incorporating targeting ligands, such as affibodies or antibodies, could enhance tumor-specific accumulation during intravenous administration, facilitating clinical translation. This addition underscores our consideration of intravenous injection as a future research direction to broaden LvP-chla’s applicability while maintaining the focus on the current study’s objectives.

Comment 4: In the discussion section, the author should add the discussion on how to modify its structure to move its absorption peak to the near-infrared window (700-900 nm), which will enhance its penetration depth for deeper imaging.

Response: We appreciate your suggestion to discuss modifying LvP-chla’s structure to shift its absorption peak to the near-infrared (NIR) window (700–900 nm) for enhanced penetration depth. This point is already addressed in the Discussion (Section 4, final paragraph), where we note that LvP-chla’s current absorption peak at 665 nm may limit imaging depth compared to NIR wavelengths and propose reconstituting LvP with bacteriochlorophyll a to shift absorption peaks further into the NIR region, leveraging LvP’s structural flexibility to bind tetrapyrroles. This revision strengthens the focus on NIR optimization as a key future direction for enhancing LvP-chla’s clinical potential.

Reviewer 2 Report

Comments and Suggestions for Authors

This paper demonstrates a new PA contrast agent LvP-Chla, which is interesting and a nice contrast agent for PA imaging. The presentation of the paper is excellent and clear. 

A discussable point in the discussion part:

The authors claim that the PA amplitude ratio in phantom study for 665nm and 685nm is not as expected to be two folds as in the ratio of absorbance, which is supported by their data. The reasons may not be as they explained, in this reviewer's opinion. Gruneisen coefficient is only related to generated acoustic wave speed and specific heat, both are determined by their samples, which are not changed under different wavelength illumination. In my opinion, this is may be due to the absorption difference. Here, they should think energy (intensity), not the amplitude for comparison. For higher absorbance, the generated acoustic wave bandwidth should be broader, which means narrower acoustic wave profile, under the same energy, the peak amplitude would be higher. This may be the reason.

Other than this, I think the whole paper is nice written.  

Author Response

Comment: This paper demonstrates a new PA contrast agent LvP-Chla, which is interesting and a nice contrast agent for PA imaging. The presentation of the paper is excellent and clear.

A discussable point in the discussion part:

The authors claim that the PA amplitude ratio in phantom study for 665nm and 685nm is not as expected to be two folds as in the ratio of absorbance, which is supported by their data. The reasons may not be as they explained, in this reviewer's opinion. Gruneisen coefficient is only related to generated acoustic wave speed and specific heat, both are determined by their samples, which are not changed under different wavelength illumination. In my opinion, this is may be due to the absorption difference. Here, they should think energy (intensity), not the amplitude for comparison. For higher absorbance, the generated acoustic wave bandwidth should be broader, which means narrower acoustic wave profile, under the same energy, the peak amplitude would be higher. This may be the reason.

Other than this, I think the whole paper is nice written.  

Response: Thank you for your insightful feedback on our manuscript. We greatly appreciate your positive comments on the novelty and clarity of our work on LvP-chla as a photoacoustic (PA) contrast agent. We have carefully considered your comment regarding the observed threefold PA signal ratio between 665 nm and 685 nm, which exceeds the expected twofold absorbance ratio.

In PAI, the bandwidth of the generated PA signal is not directly determined by the optical absorbance alone. The bandwidth of the PA signal is influenced by various factors, including the properties of the absorbing material, the characteristics of the laser pulse used for excitation, and the acoustic properties of the tissue. Therefore, following what you mentioned, these properties are nearly identical when we simply switch the wavelengths.

In particular, higher optical absorbance can contribute to the generation of a stronger PA signal. The bandwidth of the signal is more closely related to the temporal characteristics of the laser pulse and the imaging target size. Shorter laser pulses with faster rise times result in broader bandwidth signals, or an imaging target with a smaller size also results in broader bandwidth signals. However, these two should also be the same for our imaging conditions.

In any case, we would like to sincerely thank the reviewer for hypothesizing a reason why the observed ratio is more than twofold. Further studies should be carried out to elucidate the mechanism fully, but this is beyond the scope of our proof-of-concept study.